# Hierarchical Attentive Recurrent Tracking

**Adam R. Kosiorek**
Department of Engineering Science
University of Oxford
adamk@robots.ox.ac.uk

**Alex Bewley**
Department of Engineering Science
University of Oxford
bewley@robots.ox.ac.uk

**Ingmar Posner**
Department of Engineering Science
University of Oxford
ingmar@robots.ox.ac.uk

## Abstract

Class-agnostic object tracking is particularly difficult in cluttered environments as target specific discriminative models cannot be learned *a priori*. Inspired by how the human visual cortex employs spatial attention and separate "where" and "what" processing pathways to actively suppress irrelevant visual features, this work develops a hierarchical attentive recurrent model for single object tracking in videos. The first layer of attention discards the majority of background by selecting a region containing the object of interest, while the subsequent layers tune in on visual features *particular* to the tracked object. This framework is fully differentiable and can be trained in a purely data driven fashion by gradient methods. To improve training convergence, we augment the loss function with terms for auxiliary tasks relevant for tracking. Evaluation of the proposed model is performed on two datasets: pedestrian tracking on the KTH activity recognition dataset and the more difficult KITTI object tracking dataset.

## 1 Introduction

In computer vision, designing an algorithm for model-free tracking of anonymous objects is challenging, since no target-specific information can be gathered *a priori* and yet the algorithm has to handle target appearance changes, varying lighting conditions and occlusion. To make it even more difficult, the tracked object often constitutes but a small fraction of the visual field. The remaining parts may contain *distractors*, which are visually salient objects resembling the target but hold no relevant information. Despite this fact, recent models often process the whole image, which exposes them to noise and increases the associated computational cost or they use heuristic methods to decrease the size of search regions. This in contrast to human visual perception, which does not process the visual field in its entirety, but rather acknowledges it briefly and focuses on processing small fractions thereof, which we dub *visual attention*.

Attention mechanisms have recently been explored in machine learning in a wide variety of contexts [27, 14], often providing new capabilities to machine learning algorithms [11, 12, 7]. While they improve efficiency [22] and performance on state-of-the-art machine learning benchmarks [27], their architecture is much simpler than that of the mechanisms found in the human visual cortex [5]. Attention has also been long studied by neuroscientists [18], who believe that it is crucial for visual perception and cognition [4], since it is inherently tied to the architecture of the visual cortex and can affect the information flow inside it. Whenever more than one visual stimulus is present in the receptive field of a neuron, all the stimuli compete for computational resources due to the limited processing capacity. Visual attention can lead to suppression of distractors by reducing the size of

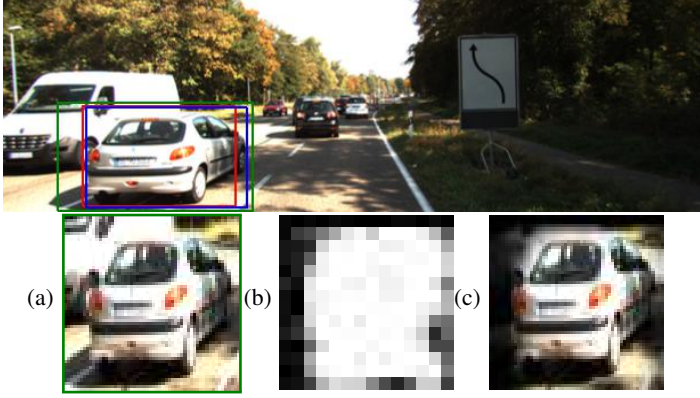

Figure 1: KITTI image with the ground-truth and predicted bounding boxes and an attention glimpse. The lower row corresponds to the hierarchical attention of our model: $1^{st}$ layer extracts an attention glimpse (a), the $2^{nd}$ layer uses appearance attention to build a location map (b). The $3^{rd}$ layer uses the location map to suppress distractors, visualised in (c).

(a)     (b)     (c)

the receptive field of a neuron and by increasing sensitivity at a given location in the visual field (*spatial attention*). It can also amplify activity in different parts of the cortex, which are specialised in processing different types of features, leading to response enhancement with respect to those features (*appearance attention*). The functional separation of the visual cortex is most apparent in two distinct processing pathways. After leaving the eye, the sensory inputs enter the primary visual cortex (known as *V1*) and then split into the *dorsal stream*, responsible for estimating spatial relationships (*where*), and the *ventral stream*, which targets appearance-based features (*what*).

Inspired by the general architecture of the human visual cortex and the role of attention mechanisms, this work presents a biologically-inspired recurrent model for single object tracking in videos (*cf.* section 3). Tracking algorithms typically use simple motion models and heuristics to decrease the size of the search region. It is interesting to see whether neuroscientific insights can aid our computational efforts, thereby improving the efficiency and performance of single object tracking. It is worth noting that visual attention can be induced by the stimulus itself (due to, e. g., high contrast) in a *bottom-up* fashion or by back-projections from other brain regions and working memory as *top-down* influence. The proposed approach exploits this property to create a feedback loop that steers the *three* layers of visual attention mechanisms in our hierarchical attentive recurrent tracking (*HART*) framework, see Figure 1. The first stage immediately discards spatially irrelevant input, while later stages focus on producing target-specific filters to emphasise visual features *particular* to the object of interest.

The resulting framework is end-to-end trainable and we resort to maximum likelihood estimation (MLE) for parameter learning. This follows from our interest in estimating the distribution over object locations in a sequence of images, given the initial location from whence our tracking commenced. Formally, given a sequence of images $\mathbf{x}_{1:T} \in \mathbb{R}^{H \times W \times C}$, where the superscript denotes height, width and the number of channels of the image, respectively, and an initial location for the tracked object given by a bounding box $\mathbf{b}_1 \in \mathbb{R}^4$, the conditional probability distribution factorises as

$$p(\mathbf{b}_{2:T} \mid \mathbf{x}_{1:T}, \mathbf{b}_1) = \int p(\boldsymbol{h}_1 \mid \mathbf{x}_1, \mathbf{b}_1) \prod_{t=2}^{T} \int p(\mathbf{b}_t \mid \boldsymbol{h}_t) p(\boldsymbol{h}_t \mid \mathbf{x}_t, \mathbf{b}_{t-1}, \boldsymbol{h}_{t-1}) \, \mathrm{d}\boldsymbol{h}_t \, \mathrm{d}\boldsymbol{h}_1, \quad (1)$$

where we assume that motion of an object can be described by a Markovian state $\boldsymbol{h}_t$. Our bounding box estimates are given by $\widehat{\mathbf{b}}_{2:T}$, found by the MLE of the model parameters. In sum, our contributions are threefold: Firstly, a hierarchy of attention mechanisms that leads to suppressing distractors and computational efficiency is introduced. Secondly, a biologically plausible combination of attention mechanisms and recurrent neural networks is presented for object tracking. Finally, our attention-based tracker is demonstrated using real-world sequences in challenging scenarios where previous recurrent attentive trackers have failed.

Next we briefly review related work (Section 2) before describing how information flows through the components of our hierarchical attention in Section 3. Section 4 details the losses applied to guide the attention. Section 5 presents experiments on KTH and KITTI datasets with comparison to related attention-based trackers. Section 6 discusses the results and intriguing properties of our framework and Section 7 concludes the work. Code and results are available online[1].

## 2 Related Work

A number of recent studies have demonstrated that visual content can be captured through a sequence of spatial glimpses or foveation [22, 12]. Such a paradigm has the intriguing property that the computational complexity is proportional to the number of steps as opposed to the image size. Furthermore, the fovea centralis in the retina of primates is structured with maximum visual acuity in the centre and decaying resolution towards the periphery, Cheung et al. [4] show that if spatial attention is capable of zooming, a regular grid sampling is sufficient. Jaderberg et al. [14] introduced the spatial transformer network (STN) which provides a fully differentiable means of transforming feature maps, conditioned on the input itself. Eslami et al. [7] use the STN as a form of attention in combination with a recurrent neural network (RNN) to sequentially locate and identify objects in an image. Moreover, Eslami et al. [7] use a latent variable to estimate the presence of additional objects, allowing the RNN to adapt the number of time-steps based on the input. Our spatial attention mechanism is based on the two dimensional Gaussian grid filters of [16] which is both fully differentiable and more biologically plausible than the STN.

Whilst focusing on a specific location has its merits, focusing on particular appearance features might be as important. A policy with feedback connections can learn to adjust filters of a convolutional neural network (CNN), thereby adapting them to features present in the current image and improving accuracy [25]. De Brabandere et al. [6] introduced dynamic filter network (DFN), where filters for a CNN are computed on-the-fly conditioned on input features, which can reduce model size without performance loss. Karl et al. [17] showed that an input-dependent state transitions can be helpful for learning latent Markovian state-space system. While not the focus of this work, we follow this concept in estimating the expected appearance of the tracked object.

In the context of single object tracking, both attention mechanisms and RNNs appear to be perfectly suited, yet their success has mostly been limited to simple monochromatic sequences with plain backgrounds [16]. Cheung [3] applied STNs [14] as attention mechanisms for real-world object tracking, but failed due to exploding gradients potentially arising from the difficulty of the data. Ning et al. [23] achieved competitive performance by using features from an object detector as inputs to a long-short memory network (LSTM), but requires processing of the whole image at each time-step. Two recent state-of-the-art trackers employ convolutional Siamese networks which can be seen as an RNN unrolled over two time-steps [13, 26]. Both methods explicitly process small search areas around the previous target position to produce a bounding box offset [13] or a correlation response map with the maximum corresponding to the target position [26]. We acknowledge the recent work[2] of Gordon et al. [10] which employ an RNN based model and use explicit cropping and warping as a form of non-differentiable spatial attention. The work presented in this paper is closest to [16] where we share a similar spatial attention mechanism which is guided through an RNN to effectively learn a motion model that spans multiple time-steps. The next section describes our additional attention mechanisms in relation to their biological counterparts.

## 3 Hierarchical Attention

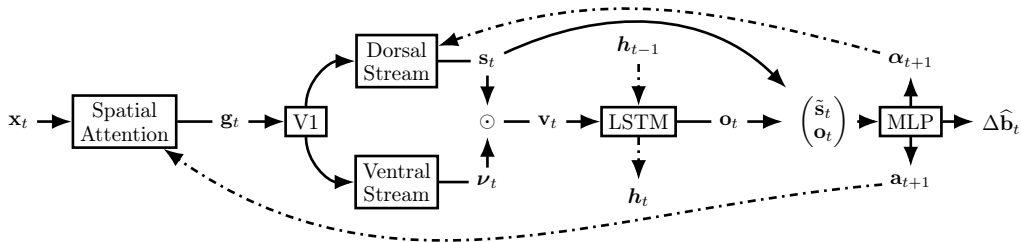

Figure 2: Hierarchical Attentive Recurrent Tracking. Spatial attention extracts a glimpse $\mathbf{g}_t$ from the input image $\mathbf{x}_t$. V1 and the ventral stream extract appearance-based features $\boldsymbol{\nu}_t$ while the dorsal stream computes a foreground/background segmentation $\mathbf{s}_t$ of the attention glimpse. Masked features $\mathbf{v}_t$ contribute to the working memory $\boldsymbol{h}_t$. The LSTM output $\mathbf{o}_t$ is then used to compute attention $\mathbf{a}_{t+1}$, appearance $\boldsymbol{\alpha}_{t+1}$ and a bounding box correction $\Delta\widehat{\mathbf{b}}_t$. Dashed lines correspond to temporal connections, while solid lines describe information flow within one time-step.

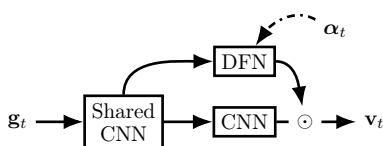

Figure 3: Architecture of the appearance attention. V1 is implemented as a CNN shared among the dorsal stream (DFN) and the ventral stream (CNN). The $\odot$ symbol represents the Hadamard product and implements masking of visual features by the foreground/background segmentation.

Inspired by the architecture of the human visual cortex, we structure our system around working memory responsible for storing the motion pattern and an appearance description of the tracked object. If both quantities were known, it would be possible to compute the expected location of the object at the next time step. Given a new frame, however, it is not immediately apparent which visual features correspond to the appearance description. If we were to pass them on to an RNN, it would have to implicitly solve a data association problem. As it is non-trivial, we prefer to model it explicitly by outsourcing the computation to a separate processing stream conditioned on the expected appearance. This results in a location-map, making it possible to neglect features inconsistent with our memory of the tracked object. We now proceed with describing the information flow in our model.

Given attention parameters $\mathbf{a}_t$, the *spatial attention* module extracts a glimpse $\mathbf{g}_t$ from the input image $\mathbf{x}_t$. We then apply *appearance attention*, parametrised by appearance $\boldsymbol{\alpha}_t$ and comprised of V1 and dorsal and ventral streams, to obtain object-specific features $\mathbf{v}_t$, which are used to update the hidden state $\boldsymbol{h}_t$ of an LSTM. The LSTM's output is then decoded to predict both spatial and appearance attention parameters for the next time-step along with a bounding box correction $\Delta \widehat{\mathbf{b}}_t$ for the current time-step. Spatial attention is driven by top-down signal $\mathbf{a}_t$, while appearance attention depends on top-down $\boldsymbol{\alpha}_t$ as well as bottom-up (contents of the glimpse $\mathbf{g}_t$) signals. Bottom-up signals have local influence and depend on stimulus salience at a given location, while top-down signals incorporate global context into local processing. This attention hierarchy, further enhanced by recurrent connections, mimics that of the human visual cortex [18]. We now describe the individual components of the system.

**Spatial Attention** Our spatial attention mechanism is similar to the one used by Kahoú et al. [16]. Given an input image $\mathbf{x}_t \in \mathbb{R}^{H \times W}$, it creates two matrices $\mathbf{A}_t^x \in \mathbb{R}^{w \times W}$ and $\mathbf{A}_t^y \in \mathbb{R}^{h \times H}$, respectively. Each matrix contains one Gaussian per row; the width and positions of the Gaussians determine which parts of the image are extracted as the attention glimpse. Formally, the glimpse $\mathbf{g}_t \in \mathbb{R}^{h \times w}$ is defined as

$$\mathbf{g}_t = \mathbf{A}_t^y \mathbf{x}_t \left(\mathbf{A}_t^x\right)^\mathsf{T}. \tag{2}$$

Attention is described by centres $\mu$ of the Gaussians, their variances $\sigma^2$ and strides $\gamma$ between centers of Gaussians of consecutive rows of the matrix, one for each axis. In contrast to the work by Kahoú et al. [16], only centres and strides are estimated from the hidden state of the LSTM, while the variance depends solely on the stride. This prevents excessive aliasing during training caused when predicting a small variance (compared to strides) leading to smoother convergence. The relationship between variance and stride is approximated using linear regression with polynomial basis functions (up to $4^{th}$ order) before training the whole system. The glimpse size we use depends on the experiment.

**Appearance Attention** This stage transforms the attention glimpse $\mathbf{g}_t$ into a fixed-dimensional vector $\mathbf{v}_t$ comprising appearance and spatial information about the tracked object. Its architecture depends on the experiment. In general, however, we implement V1 : $\mathbb{R}^{h \times w} \to \mathbb{R}^{h_v \times w_v \times c_v}$ as a number of convolutional and max-pooling layers. They are shared among later processing stages, which corresponds to the primary visual cortex in humans [5]. Processing then splits into ventral and dorsal streams. The ventral stream is implemented as a CNN, and handles visual features and outputs feature maps $\boldsymbol{\nu}_t$. The dorsal stream, implemented as a DFN, is responsible for handling spatial relationships. Let $\mathrm{MLP}(\cdot)$ denote a multi-layered perceptron. The dorsal stream uses appearance $\boldsymbol{\alpha}_t$ to dynamically compute convolutional filters $\boldsymbol{\psi}_t^{a \times b \times c \times d}$, where the superscript denotes the size of the filters and the number of input and output feature maps, as

$$\boldsymbol{\Psi}_t = \left\{ \boldsymbol{\psi}_t^{a_i \times b_i \times c_i \times d_i} \right\}_{i=1}^{K} = \mathrm{MLP}(\boldsymbol{\alpha}_t). \tag{3}$$

The filters with corresponding nonlinearities form $K$ convolutional layers applied to the output of V1. Finally, a convolutional layer with a $1 \times 1$ kernel and a sigmoid non-linearity is applied to transform the output into a spatial Bernoulli distribution $\mathbf{s}_t$. Each value in $\mathbf{s}_t$ represents the probability of the tracked object occupying the corresponding location.

The location map of the dorsal stream is combined with appearance-based features extracted by the ventral stream, to imitate the distractor-suppressing behaviour of the human brain. It also prevents drift and allows occlusion handling, since object appearance is not overwritten in the hidden state when input does not contain features particular to the tracked object. Outputs of both streams are combined as[3]

$$\mathbf{v}_t = \mathrm{MLP}(\mathrm{vec}(\boldsymbol{\nu}_t \odot \mathbf{s}_t)), \tag{4}$$

with $\odot$ being the Hadamard product.

**State Estimation** Our approach relies on being able to predict future object appearance and location, and therefore it heavily depends on state estimation. We use an LSTM, which can learn to trade-off spatio-temporal and appearance information in a data-driven fashion. It acts like a working memory, enabling the system to be robust to occlusions and oscillating object appearance e. g., when an object rotates and comes back to the original orientation.

$$\mathbf{o}_t, \boldsymbol{h}_t = \mathrm{LSTM}(\mathbf{v}_t, \boldsymbol{h}_{t-1}), \tag{5}$$

$$\boldsymbol{\alpha}_{t+1}, \Delta\mathbf{a}_{t+1}, \Delta\widehat{\mathbf{b}}_t = \mathrm{MLP}(\mathbf{o}_t, \mathrm{vec}(\mathbf{s}_t)), \tag{6}$$

$$\mathbf{a}_{t+1} = \mathbf{a}_t + \tanh(\boldsymbol{c})\Delta\mathbf{a}_{t+1}, \tag{7}$$

$$\widehat{\mathbf{b}}_t = \mathbf{a}_t + \Delta\widehat{\mathbf{b}}_t \tag{8}$$

Equations (5) to (8) detail the state updates. Spatial attention at time $t$ is formed as a cumulative sum of attention updates from times $t = 1$ to $t = T$, where $\boldsymbol{c}$ is a learnable parameter initialised to a small value to constrain the size of the updates early in training. Since the spatial-attention mechanism is trained to predict where the object is going to go (Section 4), the bounding box $\widehat{\mathbf{b}}_t$ is estimated relative to attention at time $t$.

# 4 Loss

We train our system by minimising a loss function comprised of: a tracking loss term, a set of terms for auxiliary tasks and regularisation terms. Auxiliary tasks are essential for real-world data, since convergence does not occur without them. They also speed up learning and lead to better performance for simpler datasets. Unlike the auxiliary tasks used by Jaderberg et al. [15], ours are relevant for our main objective — object tracking. In order to limit the number of hyperparameters, we automatically learn loss weighting. The loss $\mathcal{L}(\cdot)$ is given by

$$\mathcal{L}_{\mathrm{HART}}(\mathcal{D}, \theta) = \lambda_{\mathrm{t}}\mathcal{L}_{\mathrm{t}}(\mathcal{D}, \theta) + \lambda_{\mathrm{s}}\mathcal{L}_{\mathrm{s}}(\mathcal{D}, \theta) + \lambda_{\mathrm{a}}\mathcal{L}_{\mathrm{a}}(\mathcal{D}, \theta) + R(\boldsymbol{\lambda}) + \beta R(\mathcal{D}, \theta), \tag{9}$$

with dataset $\mathcal{D} = \left\{ (\mathbf{x}_{1:T}, \mathbf{b}_{1:T})^i \right\}_{i=1}^{M}$, network parameters $\theta$, regularisation terms $R(\cdot)$, adaptive weights $\boldsymbol{\lambda} = \{\lambda_{\mathrm{t}}, \lambda_{\mathrm{s}}, \lambda_{\mathrm{d}}\}$ and a regularisation weight $\beta$. We now present and justify components of our loss, where expectations $\mathbb{E}[\cdot]$ are evaluated as an empirical mean over a minibatch of samples $\left\{ \mathbf{x}_{1:T}^i, \mathbf{b}_{1:T}^i \right\}_{i=1}^{M}$, where $M$ is the batch size.

**Tracking** To achieve the main tracking objective (localising the object in the current frame), we base the first loss term on Intersection-over-Union (IoU) of the predicted bounding box w. r. t. the ground truth, where the IoU of two bounding boxes is defined as $\mathrm{IoU}(\boldsymbol{a}, \boldsymbol{b}) = \frac{\boldsymbol{a} \cap \boldsymbol{b}}{\boldsymbol{a} \cup \boldsymbol{b}} = \frac{\text{area of overlap}}{\text{area of union}}$. The IoU is invariant to object and image scale, making it a suitable proxy for measuring the quality of localisation. Even though it (or an exponential thereof) does not correspond to any probability distribution (as it cannot be normalised), it is often used for evaluation [20]. We follow the work by Yu et al. [28] and express the loss term as the negative log of IoU:

$$\mathcal{L}_{\mathrm{t}}(\mathcal{D}, \theta) = \mathbb{E}_{p(\widehat{\mathbf{b}}_{1:T} | \mathbf{x}_{1:T}, \mathbf{b}_1)} \left[ -\log \mathrm{IoU}(\widehat{\mathbf{b}}_t, \mathbf{b}_t) \right], \tag{10}$$

with IoU clipped for numerical stability.

time

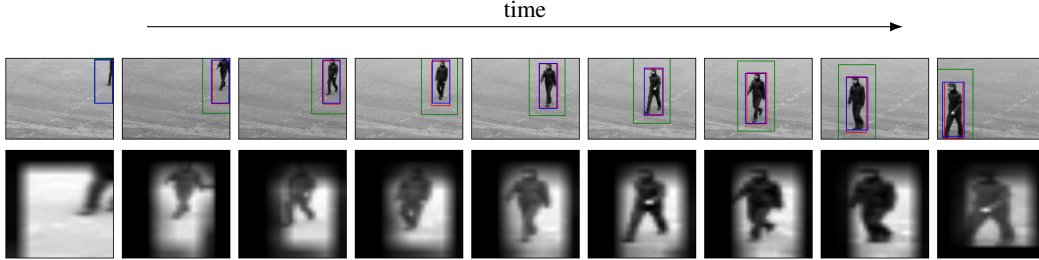

Figure 4: Tracking results on KTH dataset [24]. Starting with the first initialisation frame where all three boxes overlap exactly, time flows from left to right showing every $16^{th}$ frame of the sequence captured at 25fps. The colour coding follows from Figure 1. The second row shows attention glimpses multiplied with appearance attention.

**Spatial Attention** Spatial attention singles out the tracked object from the image. To estimate its parameters, the system has to predict the object's motion. In case of an error, especially when the attention glimpse does not contain the tracked object, it is difficult to recover. As the probability of such an event increases with decreasing size of the glimpse, we employ two loss terms. The first one constrains the predicted attention to cover the bounding box, while the second one prevents it from becoming too large, where the logarithmic arguments are appropriately clipped to avoid numerical instabilities:

$$\mathcal{L}_s(\mathcal{D}, \theta) = \mathbb{E}_{p(\mathbf{a}_{1:T}|\mathbf{x}_{1:T}, \mathbf{b}_1)}\left[ -\log\left( \frac{\mathbf{a}_t \cap \mathbf{b}_t}{\text{area}(\mathbf{b}_t)} \right) - \log(1 - \text{IoU}(\mathbf{a}_t, \mathbf{x}_t)) \right]. \tag{11}$$

**Appearance Attention** The purpose of appearance attention is to suppress distractors while keeping full view of the tracked object e.g., focus on a *particular* pedestrian moving within a group. To guide this behaviour, we put a loss on appearance attention that encourages picking out only the tracked object. Let $\tau(\mathbf{a}_t, \mathbf{b}_t) : \mathbb{R}^4 \times \mathbb{R}^4 \to \{0, 1\}^{h_v \times w_v}$ be a target function. Given the bounding box $\mathbf{b}$ and attention $\mathbf{a}$, it outputs a binary mask of the same size as the output of V1. The mask corresponds to the the glimpse $\mathbf{g}$, with the value equal to one at every location where the bounding box overlaps with the glimpse and equal to zero otherwise. If we take $H(p, q) = -\sum_z p(z) \log q(z)$ to be the cross-entropy, the loss reads

$$\mathcal{L}_a(\mathcal{D}, \theta) = \mathbb{E}_{p(\mathbf{a}_{1:T}, \mathbf{s}_{1:T}|\mathbf{x}_{1:T}, \mathbf{b}_1)}[H(\tau(\mathbf{a}_t, \mathbf{b}_t), \mathbf{s}_t)]. \tag{12}$$

**Regularisation** We apply the L2 regularisation to the model parameters $\theta$ and to the expected value of dynamic parameters $\boldsymbol{\psi}_t(\boldsymbol{\alpha}_t)$ as $R(\mathcal{D}, \theta) = \frac{1}{2}\|\theta\|_2^2 + \frac{1}{2}\|\mathbb{E}_{p(\boldsymbol{\alpha}_{1:T}|\mathbf{x}_{1:T}, \mathbf{b}_1)}[\boldsymbol{\Psi}_t \mid \boldsymbol{\alpha}_t]\|_2^2$.

**Adaptive Loss Weights** To avoid hyper-parameter tuning, we follow the work by Kendall et al. [19] and learn the loss weighting $\boldsymbol{\lambda}$. After initialising the weights with a vector of ones, we add the following regularisation term to the loss function: $R(\boldsymbol{\lambda}) = -\sum_i \log(\boldsymbol{\lambda}_i^{-1})$.

## 5 Experiments

### 5.1 KTH Pedestrian Tracking

Kahoú et al. [16] performed a pedestrian tracking experiment on the KTH activity recognition dataset [24] as a real-world case-study. We replicate this experiment for comparison. We use code provided by the authors for data preparation and we also use their pre-trained feature extractor. Unlike them, we did not need to upscale ground-truth bounding boxes by a factor of 1.5 and then downscale them again for evaluation. We follow the authors and set the glimpse size $(h, w) = (28, 28)$. We replicate the training procedure exactly, with the exception of using the RMSProp optimiser [9] with learning rate of $3.33 \times 10^{-5}$ and momentum set to 0.9 instead of the stochastic gradient descent with momentum. The original work reported an IoU of 55.03% on average, on test data, while the presented work achieves an average IoU score of 77.11%, reducing the relative error by almost a factor of two. Figure 4 presents qualitative results.

### 5.2 Scaling to Real-World Data: KITTI

Since we demonstrated that pedestrian tracking is feasible using the proposed architecture, we proceed to evaluate our model in a more challenging multi-class scenario on the KITTI dataset [8]. It consists

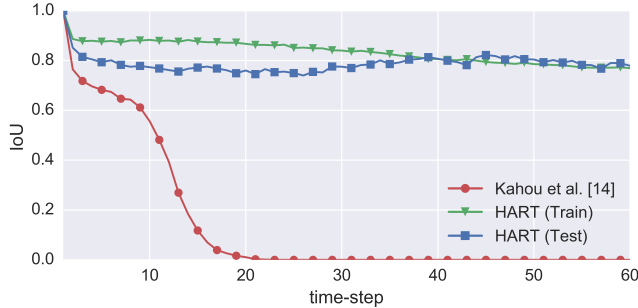

Figure 5: IoU curves on KITTI over 60 timesteps. HART (train) presents evaluation on the train set (we do not overfit).

| Method | Avg. IoU |
|---|---|
| Kahoú et al. [16] | 0.14 |
| Spatial Att | 0.60 |
| App Att | 0.78 |
| HART | **0.81** |

Table 1: Average IoU on KITTI over 60 time-steps.

of 21 high resolution video sequences with multiple instances of the same class posing as potential distractors. We split all sequences into 80/20 sequences for train and test sets, respectively. As images in this dataset are much more varied, we implement V1 as the first three convolutional layers of a modified AlexNet [1]. The original AlexNet takes inputs of size $227 \times 227$ and downsizes them to $14 \times 14$ after *conv3* layer. Since too low resolution would result in low tracking performance, and we did not want to upsample the extracted glimpse, we decided to replace the initial stride of four with one and to skip one of the max-pooling operations to conserve spatial dimensions. This way, our feature map has the size of $14 \times 14 \times 384$ with the input glimpse of size $(h, w) = (56, 56)$. We apply dropout with probability 0.25 at the end of V1. The ventral stream is comprised of a single convolutional layer with a $1 \times 1$ kernel and five output feature maps. The dorsal stream has two dynamic filter layers with kernels of size $1 \times 1$ and $3 \times 3$, respectively and five feature maps each. We used 100 hidden units in the RNN with orthogonal initialisation and Zoneout [21] with probability set to 0.05. The system was trained via curriculum learning [2], by starting with sequences of length five and increasing sequence length every 13 epochs, with epoch length decreasing with increasing sequence length. We used the same optimisation settings, with the exception of the learning rate, which we set to $3.33 \times 10^{-6}$.

Table 1 and Figure 5 contain results of different variants of our model and of the RATM tracker by Kahoú et al. [16] related works. *Spatial Att* does not use appearance attention, nor loss on attention parameters. *App Att* does not apply any loss on appearance attention, while *HART* uses all described modules; it is also our biggest model with 1.8 million parameters. Qualitative results in the form of a video with bounding boxes and attention are available online [4]. We implemented the RATM tracker of Kahoú et al. [16] and trained with the same hyperparameters as our framework, since both are closely related. It failed to learn even with the initial curriculum of five time-steps, as RATM cannot integrate the frame $\mathbf{x}_t$ into the estimate of $\mathbf{b}_t$ (it predicts location at the next time-step). Furthermore, it uses feature-space distance between ground-truth and predicted attention glimpses as the error measure, which is insufficient on a dataset with rich backgrounds. It did better when we initialised its feature extractor with weights of our trained model but, despite passing a few stags of the curriculum, it achieved very poor final performance.

## 6 Discussion

The experiments in the previous section show that it is possible to track real-world objects with a recurrent attentive tracker. While similar to the tracker by Kahoú et al. [16], our approach uses additional building blocks, specifically: (i) bounding-box regression loss, (ii) loss on spatial attention, (iii) appearance attention with an additional loss term, and (iv) combines all of these in a unified approach. We now discuss properties of these modules.

**Spatial Attention Loss prevents Vanishing Gradients** Our early experiments suggest that using only the tracking loss causes an instance of the vanishing gradient problem. Early in training, the system is not able to estimate object's motion correctly, leading to cases where the extracted glimpse does not contain the tracked object or contains only a part thereof. In such cases, the supervisory signal is only weakly correlated with the model's input, which prevents learning. Even when the object is contained within the glimpse, the gradient path from the loss function is rather long, since any teaching signal has to pass to the previous timestep through the feature extractor stage. Penalising attention parameters directly seems to solve this issue.

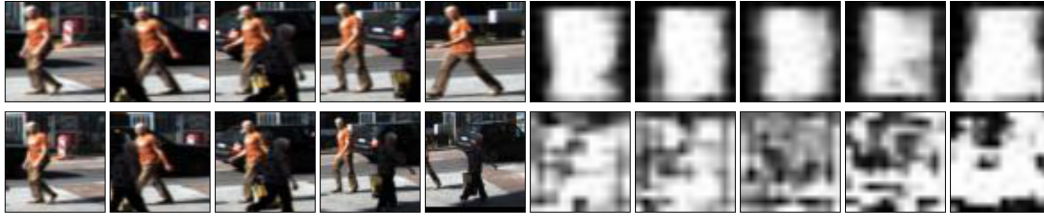

(a) The model with appearance attention loss (top) learns to focus on the tracked object, which prevents an ID swap when a pedestrian is occluded by another one (bottom).

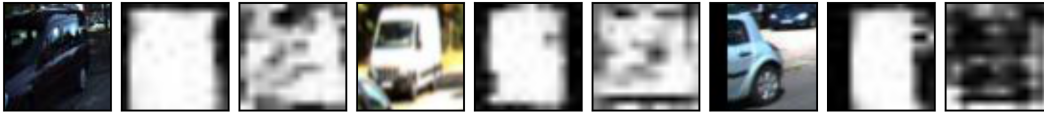

(b) Three examples of glimpses and locations maps for a model with and without appearance loss (left to right). Attention loss forces the appearance attention to pick out only the tracked object, thereby suppressing distractors.

Figure 6: Glimpses and corresponding location maps for models trained with and without appearance loss. The appearance loss encourages the model to learn foreground/background segmentation of the input glimpse.

**Is Appearance Attention Loss Necessary?** Given enough data and sufficiently high model capacity, appearance attention should be able to filter out irrelevant input features before updating the working memory. In general, however, this behaviour can be achieved faster if the model is constrained to do so by using an appropriate loss. Figure 6 shows examples of glimpses and corresponding location maps for a model with and without loss on the appearance attention. In figure 6a the model with loss on appearance attention is able to track a pedestrian even after it was occluded by another human. Figure 6b shows that, when not penalised, location map might not be very object-specific and can miss the object entirely (right-most figure). By using the appearance attention loss, we not only improve results but also make the model more interpretable.

**Spatial Attention Bias is Always Positive** To condition the system on the object's appearance and make it independent of the starting location, we translate the initial bounding box to attention parameters, to which we add a learnable bias, and create the hidden state of LSTM from corresponding visual features. In our experiments, this bias always converged to positive values favouring attention glimpse slightly larger than the object bounding box. It suggests that, while discarding irrelevant features is desirable for object tracking, the system as a whole learns to trade off attention responsibility between the spatial and appearance based attention modules.

## 7 Conclusion

Inspired by the cascaded attention mechanisms found in the human visual cortex, this work presented a neural attentive recurrent tracking architecture suited for the task of object tracking. Beyond the biological inspiration, the proposed approach has a desirable computational cost and increased interpretability due to location maps, which select features essential for tracking. Furthermore, by introducing a set of auxiliary losses we are able to scale to challenging real world data, outperforming predecessor attempts and approaching state-of-the-art performance. Future research will look into extending the proposed approach to multi-object tracking, as unlike many single object tracking, the recurrent nature of the proposed tracker offers the ability to attend each object in turn.

## Acknowledgements

We would like to thank Oiwi Parker Jones and Martin Engelcke for discussions and valuable insights and Neil Dhir for his help with editing the paper. Additionally, we would like to acknowledge the support of the UK's Engineering and Physical Sciences Research Council (EPSRC) through the Programme Grant EP/M019918/1 and the Doctoral Training Award (DTA). The donation from Nvidia of the Titan Xp GPU used in this work is also gratefully acknowledged.

## Footnotes

[1] https://github.com/akosiorek/hart

[2][10] only became available at the time of submitting this paper.

[3] $\mathrm{vec} : \mathbb{R}^{m \times n} \to \mathbb{R}^{mn}$ is the vectorisation operator, which stacks columns of a matrix into a column vector.

[4] https://youtu.be/VvkjmOFRGSs

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
