[Reviews · NeurIPS 2017]

Reviewer 1



The paper aims to track a single object in videos. The objective is to maximize the Intersection-over-Union of the predicted box with respect to the ground truth bounding box. The paper uses attention from DRAW to focus only on a subset of the screen. The paper adds extra losses to use the available ground truth boxes as attention targets during training. The network may be a good approach for tracking. I am mainly missing a comparison to the state of the art on a standard benchmark. It would be nice to see improvements on a practically important task. The paper provides a comparison only to a method designed for a different objective (next step prediction). As expected, looking at the current frame improves the produced bounding box for the current frame. If looking at future frames is allowed, another method can look at multiple future frames. The dataset with 21 videos seems too small. If training on a bigger dataset, a less regularized net may be better. Clarification questions: - How are the logarithmic arguments clipped? Is the gradient zero, if a value is clipped? Typos: - Line 38: s/prefrontal/primary visual/ - Equation (1) has two = =. - Line 67: ImageNet video dataset is not present in experiments. - Line 172: s/a:/: a/ - Footnote 2: s/vecorisation/vectorisation/ - Citation [14]: The paper title should be "RATM: Recurrent Attentive Tracking Model". Update: I have read the rebuttal. It would be good to use a standard benchmark or define a new reproducible benchmark.

Reviewer 2



Summary: The authors present a framework for class-agnostic object tracking. In each time step, the presented model extracts a glimpse from the current frame using the DRAW-style attention mechanism also employed by Kahoue et al. Low level features are then extracted from this glimpse and fed into a two-stream architecture, which is inspired by the two-stream hypothesis. The dorsal stream constructs convolutional filters, which are used to extract a spatial Bernoulli distribution, each element expressing the probability of the object occupying the corresponding locations in the feature map. This location map is then combined with the appearance-based feature map computed by the ventral stream, removing the distracting features, which are not considered to be part of the object. An LSTM is used for tracking the state of the object, with an MLP transforming the output of the LSTM into parameters for both attention mechanisms and an offset to the bounding box parameters in the previous time step. The results on KTH and the more challenging KITTI dataset are significantly closer to the state of the art than previous attention based approaches to tracking. Qualitative Assessment: The paper reads well and is easy to follow. The description of the model is detailed enough for replication and experiments are performed providing evidence for some of the premises stated in the description of the model. The results are impressive compared to previous work on attention based tracking. Since, you used a standard tracking benchmark, I think more performance numbers from the tracking community could have been included to show how close the presented method is to the state of the art. (example: Re3: Real-Time Recurrent Regression Networks for Object Tracking D Gordon, A Farhadi, D Fox, arXiv preprint arXiv:1705.06368)

Reviewer 3



The paper presents a single object tracking framework based on attention mechanisms. The key insights are (1) learning spatial attention glimpse for coarse search and (2) using appearance attention to extract features without distractors. The whole pipeline incorporated with recurrent connections is novel for exploiting both motion and appearance information. Each part of the framework is reasonable and well designed. The spatial attention and appearnce attention parts are hierarchically stacked and incorporate global context with top-down signals. All the components are integrated into one compact framwork and can be trained end-to-end. As pointed in the abstract and intraduction, target-specific model cannot be learnt a priori. From what I understand, the CNN and DFN in the proposed method are used for learning target-specific apperance feature, and the spatial attention model exploits the class-agnostic information. The insights for each part in the framework are well explained, but it's hard to follow the motivation of the whole story. The experiments and analysis are sufficient and favorable. It achieves significant improvements with meaningful visual results. I tend to vote for an acceptance.